# Energetic-Materials-Driven Synthesis of Graphene-Encapsulated Tin Oxide Nanoparticles for Sodium-Ion Batteries

**DOI:** 10.3390/ma14102550

**Published:** 2021-05-14

**Authors:** Yingchun Wang, Jinxu Liu, Min Yang, Lijuan Hou, Tingting Xu, Shukui Li, Zhihua Zhuang, Chuan He

**Affiliations:** 1School of Materials Science and Engineering, Beijing Institute of Technology, Beijing 100081, China; 18361232309@163.com (Y.W.); liujinxu@bit.edu.cn (J.L.); 15326716870@163.com (M.Y.); bitleesk@bit.edu.cn (S.L.); zhihua0802@163.com (Z.Z.); 2Key Laboratory of Materials Physics of Ministry of Education, Zhengzhou University, Zhengzhou 450052, China; houlijuan0815@163.com (L.H.); xutt@zzu.edu.cn (T.X.); 3Department of Materials Science and Engineering, Shenzhen MSU-BIT University, Shenzhen 518172, China

**Keywords:** sodium-ion batteries, graphene, core-shell structure, energetic materials, tin oxide, nanomaterials

## Abstract

By evenly mixing polytetrafluoroethylene-silicon energetic materials (PTFE-Si EMs) with tin oxide (SnO_2_) particles, we demonstrate a direct synthesis of graphene-encapsulated SnO_2_ (Gr-SnO_2_) nanoparticles through the self-propagated exothermic reaction of the EMs. The highly exothermic reaction of the PTFE-Si EMs released a huge amount of heat that induced an instantaneous temperature rise at the reaction zone, and the rapid expansion of the gaseous SiF_4_ product provided a high-speed gas flow for dispersing the molten particles into finer nanoscale particles. Furthermore, the reaction of the PTFE-NPs with Si resulted in a simultaneous synthesis of graphene that encapsulated the SnO_2_ nanoparticles in order to form the core-shell nanostructure. As sodium storage material, the graphene-encapsulated SnO_2_ nanoparticles exhibit a good cycling performance, superior rate capability, and a high initial Coulombic efficiency of 85.3%. This proves the effectiveness of our approach for the scalable synthesis of core-shell-structured graphene-encapsulated nanomaterials.

## 1. Introduction

Possessing synergetic properties of dissimilar materials, carbon-encapsulated materials with a core-shell structure have been reported to exhibit excellent performance in energy storage, environmental applications, and biological applications [1,2,3]. Among these applications, carbon-encapsulated materials are mostly utilized in energy storage devices, especially in secondary batteries, such as lithium-(Li-) and sodium-ion (Na-ion) batteries, as anode materials [4,5,6,7,8]. It has been proven that the combination of high-capacity anode materials with conductive carbon coating simultaneously alleviates the cracks and fractures of the active materials and increases the electrical conductivity of the electrodes, hence greatly improving the performance of the anodes. As for core and shell materials, research studies have demonstrated that the utilization of nanoscale active materials in the inner core can efficiently prevent structural fracture [9,10,11,12] and that using graphene as the outer shell can significantly improve the initial Coulombic efficiency compared to its carbonaceous counterparts [12,13]. In this regard, intensive efforts have been made toward the synthesis of graphene-encapsulated nanomaterials for secondary battery anode materials [14,15,16]. However, most of the technological advances involve complex multistep processing and the use of commercial nanomaterials, thus limiting their further application in anode materials as a result of high costs [17].

In recent years, the synthesis of nanomaterials at high temperatures has drawn great attention for its low-energy consumption, relatively low cost, and scalability [18,19,20,21]. In our previous study, we have demonstrated the production of graphene-encapsulated Si nanoparticles through the intense exothermic reaction of polytetrafluoroethylene-silicon (PTFE-Si) energetic materials (EMs) [22]. As anode material in Li-ion batteries, the core-shell-structured graphene-Si nanocomposites exhibited a good cycling performance, proving the effectiveness of the exothermic reaction for the synthesis of graphene-encapsulated nanomaterials. In this case, Si served a dual role as the metal fuel, which reacted with the PTFE to release heat, and as the Si nanoparticles left after the reaction as the active materials for the anode of Li-ion batteries. However, the majority of high-capacity anode materials for Li- and Na-ion batteries cannot react exothermically with the PTFE, which largely restricts the choices for the materials selection.

In this article, we demonstrate the energetic-materials-driven synthesis of graphene-encapsulated tin oxide (Gr-SnO_2_) nanoparticles. Here, the SnO_2_ is chosen for its superior storage capacity in secondary batteries, especially in Na-ion batteries (1378 mAh·g^−1^ for sodium storage), which is of particular interest due to the low toxicity and wide availability of Na [23,24]. The PTFE-Si EMs are used and evenly mixed with commercial SnO_2_ particles. Upon ignition, the intense heat released by the self-sustained solid-state exothermic reaction of the PTFE-Si EMs melts the SnO_2_ particles, and the high-pressure SiF_4_ gas produced breaks the molten SnO_2_ into smaller spherical shaped nanoparticles. Concomitantly, the reaction results in a bottom-up synthesis of graphene that encapsulates the SnO_2_ nanoparticles. Used as the anode material in Na-ion batteries, the Gr-SnO_2_ nanoparticles exhibit a long-term cyclability and superior rate capability. This demonstrates the competence of our approach for the synthesis of graphene-encapsulated nanoparticles.

## 2. Experimental Section

### 2.1. Synthesis of Gr-SnO_2_ Nanoparticles

First, 20 g PTFE microparticles were immersed into the 400 mL analytical grade ethanol. The PTFE microparticles have an average diameter of ~60 µm (Shenyang Micro Powder Factory Co., Ltd., Shenyang, China). Second, the PTFE microparticles were disintegrated into nanometer-thick PTFE nanoparticles through ultrasonic processing. The ultrasonic processing was performed for ~4 h at a frequency of 20 kHz and a power of 2.4 kW. Third, 10 g of commercial Si particles (average diameter of ~800 nm, Beijing Xingrongyuan Co., Ltd., Beijing, China) and 7.5 g of SnO_2_ particles (Aladdin, see Appendix A) were added into the solution. Next, the sonication processing was continued for ~1 h at a power of 1 kW to mix the three components evenly. Finally, the PTFE/Si/SnO_2_ mixture was obtained after the filtration and drying processes.

The PTFE/Si/SnO_2_ mixture (1 g) was placed in a metal groove (80 × 20 × 10 mm), and the exothermic reaction of the mixture was triggered using an electrically heated wire (~0.16 mm Nickel-chromium alloy wire) at one end of the groove. After passing a current of 3 A through the wire, the reaction occurred and a huge amount of soot was produced. The black powders were collected after the reaction and then magnetically stirred in the NaOH solution (1 mol/L) for ~6 h to remove the remaining Si particles. Last, centrifugation and then drying was conducted in a vacuum oven for ~12 h to obtain the Gr-SnO_2_ nanoparticles.

### 2.2. Characterization

The structure of the Gr-SnO_2_ nanoparticles was characterized by X-ray diffraction (XRD) (Rigaku, Japan) (D-8 Advance, Bruker Inc., Rigaku, Japan, 40 kV, 150 mA, Cu Kα radiation, λ = 1.5406 Å). The X-ray photoelectron spectroscopy (XPS) (ThermoFisher, Waltham, MA, USA) analyses were performed with a Thermo ESCALAB 250 Xi spectrometer instrument (monochromatic Kα X-rays at 1486.6 eV). An aluminum anode was used as the source. An Invia/Reflex Laser Micro-Raman spectroscope (Renishaw-invia) was utilized to carry out the Raman measurements (Renishaw, Wotton-under-Edge, Britain). The excitation laser beam wavelength used was 514 nm. The images of the Gr-SnO_2_ nanoparticles were obtained using Scanning electron microscopy (SEM) (GeminiSEM 300) (ZEISS, Oberkochen, Germany). Transmission electron microscopy (TEM) (JEOL, Tokyo, Japan) was carried out using JEM-2100F. High-angle annular dark-field scanning TEM (HAADF-STEM) images were obtained using an FEI Titan G2 microscope (FEI, Hillsboro, OR, USA), where an aberration corrector was equipped and a Bruker Super-X EDS detector operated at 300 kV. The in situ TEM (ZepTools, Tongling, China) characterization of the sodiation processes of the Gr-SnO_2_ nanoparticles was performed using JEM 2100 TEM operated at 200 kV.

### 2.3. Electrochemical Measurements

Using a sodium counter/reference electrode (Lizhiyuan, Taiyuan, China), a CR2025 type coin cell (Lizhiyuan, Taiyuan, China) was assembled in an Ar-filled glove box to evaluate the electrochemical performance of the sample. By mixing active material (Gr-SnO_2_ nanoparticles) in a ratio of 80:10:10 with polyvinylidene fluoride (PVDF) binder and conductive super P in 1-methyl-2-pyrrolidinone (NMP) solvent, the working electrode slurry was prepared and then coated on the copper foil and dried in a vacuum at 60 °C for 12 h. The glass fiber membrane was used as the separators. The galvanostatic charge/discharge tests were carried out on a LAND CT-2001A cell test system between 0.01 to 3 V vs. Na^+^/Na at room temperature. Cyclic Voltammetry (CV) was carried out on an IM600e electrochemical workstation (CH Instruments Ins, Shanghai, China).

## 3. Result and Discussion

The efficient production of Gr-SnO_2_ nanoparticles highly depends on the mixing uniformity of the PTFE/Si/SnO_2_ mixture. The high uniformity greatly enhances the reaction efficiency of the PTFE/Si EMs and also leads to the homogeneous heating of the SnO_2_ particles, which is of importance for the synthesis of uniform nanoparticles [25]. Thus, a solution-processing technique is employed for mixing the powders. First, the PTFE microparticles were processed in ethanol solution using high-power ultrasonic waves to form a stable colloid dispersion consisting of nanometer-thick PTFE-NPs. In our previous work, by using the PTFE-NPs as the solid-state carbon source, we have demonstrated the successful synthesis of graphene nanosheets and graphene-encapsulated Si nanoparticles through the intense exothermic reaction of PTFE-based energetic materials [22,26]. Second, the Si particles, which had an average size of ~800 nm, and the SnO_2_ particles, which had an average size of 190 ± 10 nm, were added into the solution, and the sonication processing continued for ~1 h to evenly mix the three different powders. After the drying process, the PTFE/Si/SnO_2_ mixture was obtained as the starting materials for producing the Gr-SnO_2_ nanoparticles. For the synthesis of Gr-SnO_2_ nanoparticles, the mixture was placed in a metal groove and ignited at one end of the groove in an ambient atmosphere using an electrically heated metal filament. Once ignited, self-sustained combustion occurred and a huge amount of soot was produced (see Appendix A). After the removal of residual Si in the reaction products, the Gr-SnO_2_ nanoparticles were obtained.

The TEM image of a typical Gr-SnO_2_ nanoparticle and the selected area’s electron diffraction (SAED) pattern are presented in Figure 1a(i,ii), respectively. In Figure 1a(i), it can be observed that the Gr-SnO_2_ nanoparticle has a near-spherical shape and exhibits an obvious core-shell structure. The lattice fringes in the inner core have a spacing of 0.27 ± 0.01 nm, which corresponds to the (101) planes of SnO_2_. It can also be seen that few-layer graphene with an interlayer spacing of 0.39 ± 0.02 nm coated conformally on the SnO_2_ nanoparticle to construct the core-shell structure. The SAED pattern corroborated the crystalline structure of the atomized SnO_2_ nanoparticles. The core-shell structure of the Gr-SnO_2_ nanoparticles is further verified by the elemental analysis using HAADF-STEM, as shown in Figure 1b. Besides, free-standing graphene nanosheets, including single- and few-layer graphene, are also observed (see Figure 1c). The particle size distribution of the Gr-SnO_2_ nanoparticles is characterized by TEM, and the histogram of the particle size is presented in Figure 1d, where the mean particle size is determined to be 29.9 ± 0.3 nm. The representative TEM images of the Gr-SnO_2_ nanoparticles are shown in the inset of Figure 1d. The reduction of the particle size and the observation of the graphene outer layer demonstrate the formation of core-shell nanostructures, with the SnO_2_ nanoparticles as the core and the graphene as the shell.

The XRD pattern of the Gr-SnO_2_ nanoparticles is compared with that of commercial SnO_2_ particles in Figure 2a. Clearly, after the combustion process, the crystalline structure of the SnO_2_ nanoparticles was well-maintained, with all the diffraction peaks being indexed to the tetragonal rutile structure of SnO_2_. Besides, the byproducts of the reaction, silicon carbide (SiC), are also observed in Figure 2a,b, which presents the Raman spectrum of the Gr-SnO_2_ nanoparticles. As can be seen, the peaks located at ~1343 cm^−1^, ~1589 cm^−1^, ~2691 cm^−1^, and ~2926 cm^−1^ correspond to the D, G, 2D, and D+G bands of graphene, respectively [27,28]. The absence of the characteristic bands of SnO_2_ can be ascribed to the highly intense graphitic peaks of graphene, which greatly suppress the peaks of SnO_2_ [29]. The Gr-SnO_2_ nanoparticles were also characterized by XPS. In Figure 2c, the peaks located at 486.5 eV, 495.0 eV, and 531.6 eV can be assigned to Sn 3*d*_5/2_, Sn 3*d*_3/2_, and O 1*s*, respectively, which agrees with the reported data for SnO_2_ [30,31,32]. The C 1*s* spectrum of the Gr-SnO_2_ nanoparticles is presented in Figure 2d. The peak observed at 283.1 eV corresponds to the SiC, while the peaks at 284.5, 285.3, and 288.5 eV correspond to the C-C, C-O, and −C=O groups, which indicate that the graphene produced is slightly oxidized [22,26]. Furthermore, no peak corresponding to the Si^0^ (see Appendix A) is seen, demonstrating the complete removal of the unreacted Si particles.

The formation of the core-shell-structured Gr-SnO_2_ nanoparticles can be explained and interpreted as the atomization and encapsulation processes driven by the intense exothermic reaction of PTFE-Si EMs. During this fast reaction process, which is generally in the range of hundreds of microseconds, the highly exothermic reaction of the PTFE-Si EMs releases a huge amount of heat and gaseous SiF_4_ product [33]. For the Si/PTFE mixture with a similar composition, it has been reported that the average combustion temperature is in the range of 1708 to 1889 K [34], which is comparable to the melting point of SnO_2_ (1630 ℃); hence, the released heat melted the SnO_2_ particles and induced an instantaneous temperature rise at the reaction zone. This temperature rise caused a rapid expansion of the SiF_4_ gas and the subsequent pressure rise. The pressurization rate experiments were carried out using a special-designed vessel with a 220 mL internal volume, and the detail of the experimental setup can be found elsewhere [35]. The pressurization rate of the PTFE/Si EMs of a similar composition was measured to be 5.75 MPa/s, and the peak pressure was determined to be 0.545 MPa (see Appendix A). Hence, the rapid expansion of the SiF_4_ gas provides a high-speed gas flow for dispersing the molten particles into finer nanoscale particles. Furthermore, the reaction of the PTFE-NPs with Si resulted in a synthesis of graphene, including the graphene outer layers that encapsulated the SnO_2_ nanoparticles and free-standing single- and few-layer graphene nanosheets. This is corroborated with our previous studies [22,26]. According to the above discussion, the energetic-materials-driven atomization and encapsulation processes for the production of Gr-SnO_2_ nanoparticles are schematically illustrated in Figure 3.

The capability of the Gr-SnO_2_ nanoparticles in energy storage applications was further evaluated as anode materials in Na-ion batteries. As shown in Figure 4a, assembled in a half-cell configuration, the cycling performance of the Gr-SnO_2_ nanoparticles was evaluated using deep galvanostatic cycling at a current density of 200 mA·g^−1^ between 0.01 and 3.00 V. It can be seen that the Gr-SnO_2_ electrode delivers a high discharge capacity of 469 mAh·g^−1^ at the 100th cycle. Figure 4b shows the typical galvanostatic charge/discharge profiles of the Gr-SnO_2_ nanoparticles for the 1st, 2nd, and 3rd cycles tested at a current density of 200 mA·g^−1^. The core-shell-structured nanoparticles deliver an initial discharge/charge capacity of 1230.0/1048.8 mAh·g^−1^ and an initial-cycle Coulombic efficiency (ICE) of 85.3%. As the number of cycles increases, the Coulomb efficiency gradually increases to about 100% due to the formation of a relatively stable solid electrolyte interphase (SEI) [22]. The sodiation reaction is also investigated by in situ TEM technique. Figure 4c presents the TEM high-resolution images of the Gr-SnO_2_ nanoparticles and the corresponding SAED pattern before and after the first sodiation cycle. The second SAED pattern confirms the existence of crystalline Na_15_Sn_4_ phases after the sodiation, indicating that the SnO_2_ nanoparticles in the inner core were fully sodiated [24,36].

Figure 5a presents the first CV curves of the Gr-SnO_2_ nanoparticles with a scan rate of 0.1 mV·s^−1^ in a potential window of 3 to 0.01 V. In the cathodic scan, the peak observed at around 1.0 V may originate from the formation of SEI layers and the conversion of SnO_2_ to Sn. The peak that appears at around 0.7 V can be assigned to the alloying reaction between Sn and Na ions. The characteristic peak at 0.01 V can be associated with the intercalation reaction of carbon with Na ions [37,38]. In the anodic scan, the peaks at around 0.25 V and 0.6 V correspond to the dealloying reaction of Na_15_Sn_4_. In addition, the broad peak at around 1.25 V corresponds to the conversion reaction of Sn to SnO_2_ [37]. Figure 5b shows the rate capability of Gr-SnO_2_ nanoparticles at diverse current densities ranging from 0.1 A·g^−1^ to 1 A·g^−1^, respectively. The Gr-SnO_2_ nanoparticles deliver discharge capacities of 444.9, 378.8, 304.8, 280.8, and 209.5 mAh·g^−1^ at current densities of 0.1, 0.2, 0.4, 0.5, and 1 A·g^−1^, respectively. Even after cycling at a current density of 1 A·g^−1^, the reversible capacity can still return to 448.1 mAh·g^−1^ at a current density of 0.1 A·g^−1^, which demonstrates a superior cycle stability. At a higher rate, the coated tin dioxide materials show a better mechanical adjustment ability, thus achieving a good rate performance [39].

As a critical parameter of the energy density of Na-ion full batteries, the ICE of the Gr-SnO_2_ nanoparticles is amongst the highest reported for the anode materials of Na-ion batteries [37,40,41,42,43,44,45,46]. The high ICE can be ascribed to the structural features of the Gr-SnO_2_ nanoparticles, where the nanoscale core material shortens the ion diffusion path and the graphene coating facilitates the transport of electrons and ions to the core material [13,47]. Additionally, the outer layer graphene also exhibits a good mechanical stability that supports the formation of a stable SEI layer during cycling [13]. Table 1 compares the electrochemical performance of the Gr-SnO_2_ nanoparticles with that of the reported SnO_2_-based composites. It can be found that the electrochemical performance of the Gr-SnO_2_ nanoparticles is one of the best among the SnO_2_-based composites.

## 4. Conclusions

In summary, we have presented a scalable approach for the synthesis of graphene-encapsulated SnO_2_ nanoparticles as sodium storage materials. Using a solution-processing technique, commercial SnO_2_ particles are evenly mixed with PTFE-Si EMs consisting of nanometer-thick PTFE and Si particles. Upon ignition, the intense heat and gaseous SiF_4_ product was released, leading to the atomization of the SnO_2_ particles, and the synthesis of graphene at a high temperature resulted in a simultaneous encapsulation process in order to form core-shell-structured graphene-SnO_2_ nanoparticles. Used as anode materials in Na-ion batteries, the graphene-encapsulated SnO_2_ nanoparticles exhibit a good cycling performance, rate capability, and high ICE, demonstrating the structural integrity of the Gr-SnO_2_ nanoparticle. Our approach provides a direct way to synthesize graphene-encapsulated nanomaterials through the highly exothermic reaction of EMs.

## Figures and Tables

**Figure 1 materials-14-02550-f001:**
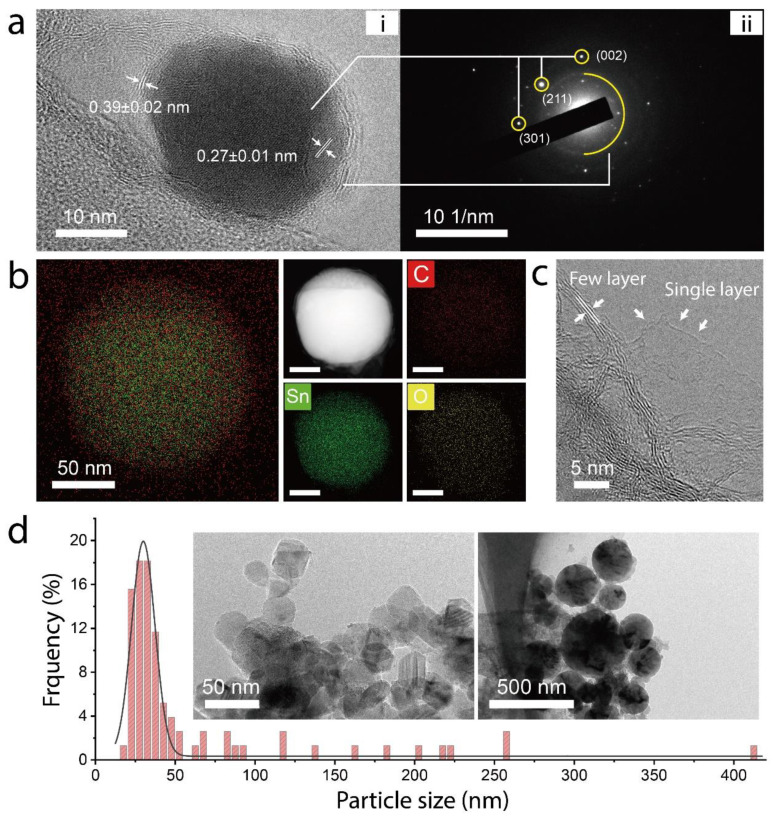
(**a**) i: The high-resolution TEM image of the Gr-SnO_2_ nanoparticle and ii: the selected area’s electron diffraction pattern corresponding to crystalline SnO_2_. The interplanar spacing of the SnO_2_ core and graphene outer layer is determined to be 0.27 ± 0.01 and 0.39 ± 0.02 nm. (**b**) The HAADF-STEM image along with the elemental analysis of the Gr-SnO_2_ nanoparticle. (**c**) The TEM images of single-layer and few-layer free-standing graphene nanosheets. (**d**) The histogram of the particle sizes of the Gr-SnO_2_ nanoparticles. The insets are the TEM images of the Gr-SnO_2_ nanoparticles.

**Figure 2 materials-14-02550-f002:**
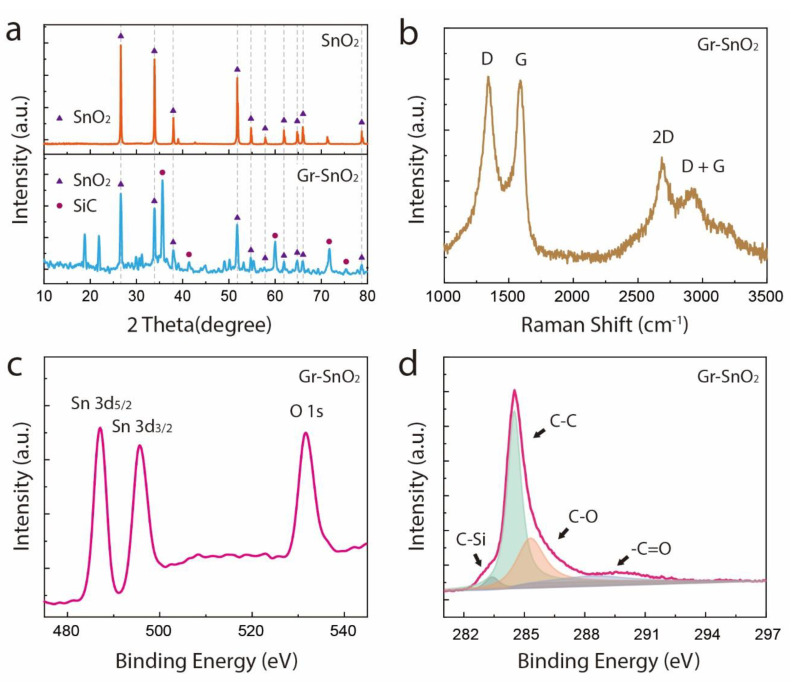
(**a**) The XRD patterns of the commercial SnO_2_ particles and Gr-SnO_2_ nanoparticles. (**b**) The Raman spectrum of the Gr-SnO_2_ nanoparticles. (**c**) The high-resolution Sn 3*d*_5/2_, Sn 3*d*_3/2_, and O 1*s* spectrum of the Gr-SnO_2_ nanoparticles. (**d**) The high-resolution C 1*s* spectrum of the Gr-SnO_2_ nanoparticles.

**Figure 3 materials-14-02550-f003:**
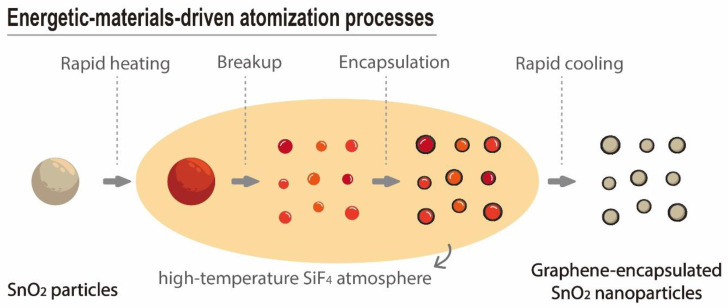
The schematic illustration of energetic-materials-driven atomization and encapsulation processes for the synthesis of graphene-encapsulated SnO_2_ nanoparticles.

**Figure 4 materials-14-02550-f004:**
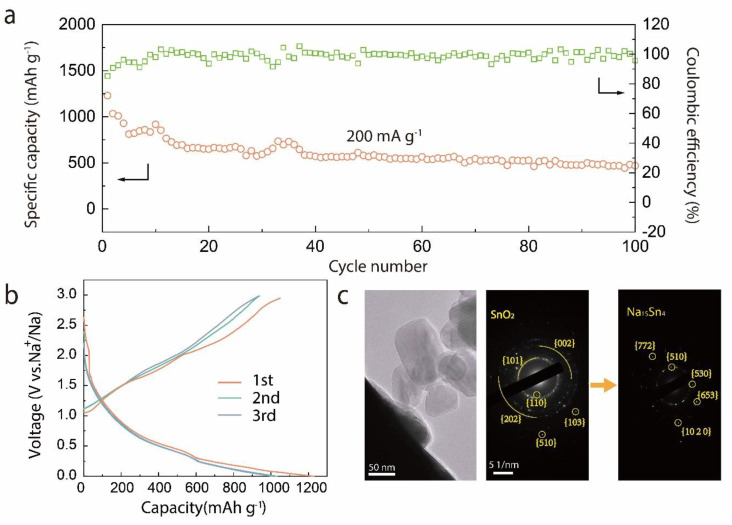
(**a**) Discharge capacity of the Gr-SnO_2_ nanoparticles for the 100 galvanostatic cycles. The current density of 200 mA·g^−1^ is used for all the cycles. (**b**) Galvanostatic charge/discharge curves of Gr-SnO_2_ nanoparticles at 200 mA·g^−1^. (**c**) The HRTEM images of the Gr-SnO_2_ nanoparticles and the SAED pattern of the unsodiated and sodiated Gr-SnO_2_ nanoparticles.

**Figure 5 materials-14-02550-f005:**
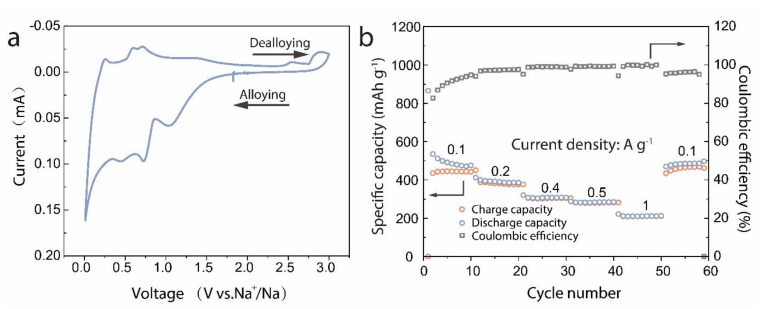
(**a**) CV curves of the Gr-SnO_2_ nanoparticles. The voltage window is 3–0.01 V. (**b**) Rate performance of the Gr-SnO_2_ nanoparticles. Current densities of 0.1, 0.2, 0.4, 0.5, and 1 A·g^−1^ are used for the cycles.

**Table 1 materials-14-02550-t001:** Comparison of the electrochemical performances of the SnO_2_-based materials for SIBs.

Materials	Cycling Performance	Rate Capacity	Coulomb Efficiency	Ref.
SnO_2_ NRs@GA	232 mAh·g^−1^ @50 mA·g^−1^	96 mAh·g^−1^ @1 A·g^−1^	58.4%	[42]
NC@SnO_2_	270 mAh·g^−1^ @100 mA·g^−1^	193 mAh·g^−1^ @1 A·g^−1^	38.2%	[43]
SnO_2_ QDs/GA	319 mAh·g^−1^ @50 mA·g^−1^	150 mAh·g^−1^ @800 mA·g^−1^	54%	[46]
CNT@SnO_2_@G	323 mAh·g^−1^ @25 mA·g^−1^	119 mAh·g^−1^ @1 A g^−1^	43%	[45]
PCS@SnO_2_@C	326 mAh·g^−1^ @50 mA·g^−1^	82 mAh·g^−1^ @1.6 A·g^−1^	53.5%	[44]
SnO_2_/NC-2	342.2 mAh·g^−1^ @100 mA·g^−1^	212.6 mAh·g^−1^ @1 A·g^−1^	59.2%	[41]
C/SnO_2_/C	370 mAh·g^−1^ @100 mA·g^−1^	105 mAh·g^−1^ @10 A·g^−1^	-	[37]
**Gr-SnO** _**2**_	**469 mAh·g** ^**−1**^ **@100 mA·g** ^**−1**^	**209.5 mAh·g** ^**−1**^ **@1 A·g** ^**−1**^	**85.3%**	**This work**

## Data Availability

The data presented in this study are available on request from the corresponding author.

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
