# Peer review of "Energetic-Materials-Driven Synthesis of Graphene-Encapsulated Tin Oxide Nanoparticles for Sodium-Ion Batteries"

_materials, 2021, doi:10.3390/ma14102550_

Round 1

Reviewer 1 Report

The manuscript “Energetic-materials-driven synthesis of graphene-encapsulated tin oxide nanoparticles for sodium-ion batteries” deals with the production of graphene-encapsulated SnO2 by a self-sustained exothermic reaction. Operating in this way, nanoparticles with a core-shell nanostructure were obtained. The work is well written and organized; intriguing results were obtained.

Detailed comments:

- Introduction. The state of the art concerning the use of graphene oxide encapsulated in porous devices can be enlarged, adding some innovative works in the field, such as the study of Sarno et al., SC-CO2-assisted process for a high energy density aerogel supercapacitor: The effect of GO loading, Nanotechnology, 2017, 28, Article number 204001; etc...

- Results. The control of the exothermic reaction should be the critical step of the process since it occurs in the range of hundreds of microseconds and at high energy. This aspect should be discussed furtherly, as well as the reproducibility of the results.

- Please, change ml with mL.

- Figures are at low resolution; please, improve the quality.

- References are not in the Journal style.

Reviewer 2 Report

The manuscript, “Energetic-materials-driven synthesis of graphene-encapsulated tin oxide nanoparticles for sodium-ion batteries” reports an interesting technique to synthesize nanosized, graphene-coated SnO­2 nanoparticles. With the recent emergence of high entropy alloys and metal oxides, this manuscript reports a fairly simple method to synthesize transition metal oxide nanoparticles for electrochemical applications. I believe this manuscript would be suitable for publication in Materials, but there are a few minor comments that need to be addressed before it could be published. The comments are as follows:

  1. Where was the combustion synthesis carried out? Was there a special vessel (or) enclosure used for the combustion reaction? Is it the same as the vessel used in “pressurization rate experiments” that the authors describe in Page 5? With the huge amount of flame and SiF4 generated, I think it is important to elaborate on the experimental setup for the sake of researchers who might want to reproduce or build upon this work.
  2. Page 6, Line 211: The authors report an average Coulombic efficiency of 100 +/- 7%. In my opinion, this value does not make much sense. It would be more meaningful to report only the first cycle CE and explain why the CEs tends to increase with cycle number, and also hypothesize why the CE values jump to more than 100% in certain cycle numbers.
  3. The authors claim that they were able to get sodiated Gr-SnO2 particles upon discharging. However, while discussing the CV (or) voltage profile results, the authors would need to explain the exact redox process and the reversible conversion reactions in Gr-SnO2 that happen during discharge and charge reactions.
  4. In Page 8, Table 1, could the authors explain how they arrived at the Coulombic efficiency value of 85.3%? I find it hard to relate this value with the cycling plots shown in the manuscript.
  5. In Page 2, Line 51. “In this case, Si severed a dual role as…”. Here, did the authors mean to say “served” instead of “severed”? I think the language could be improved throughout the manuscript and a spell-check needs to be run.

Round 2

Reviewer 1 Report

The authors answered to all questions proposed by the Reviewer and improved the manuscript.